

# Empirical and model-based estimates of spatial and temporal variations in net primary productivity in semi-arid grasslands of Northern China

Shengwei Zhang[1,2], Rui Zhang[1], Tingxi Liu[1*], Xin Song[2], Mark A. Adams[2]

[1]College of Water Conservancy and Civil Engineering, Inner Mongolia Agricultural University, Hohhot, 010018, China
[2]Faculty of Agriculture and Environment, University of Sydney, Sydney, 2570, Australia

*Correspondence to*: T. X. Liu (txliu1966@163.com) and S.W. Zhang(zsw@imau.edu.cn)

**Abstract.** Spatiotemporal variations in net primary productivity (NPP) of vegetation offer insights to surface water and carbon dynamics, and are closely related to temperature and precipitation. We employed the Carnegie-Ames-Stanford Approach ecosystem model to estimate NPP of semiarid grassland in northern China between 2001 and 2013. Model estimates were strongly linearly correlated with observed values ($R^2$=0.67, RMSE=35 g C·m$^{-2}$·year$^{-1}$). We also quantified inter-annual changes in NPP over the 13-year study period. NPP varied between 141 and 313 g C·m$^{-2}$·year$^{-1}$, with a mean of 240 g C·m$^{-2}$·year$^{-1}$. NPP increased from west to east each year, and mean precipitation in each county was significantly positively correlated with NPP in annually, summer and autumn. Mean precipitation was also positively correlated with NPP in spring, but the correlation was not significant. Annual and summer temperatures were mostly negatively correlated with NPP, but temperature was positively correlated with spring and autumn NPP. Spatial correlation and partial correlation analyses at the pixel scale confirmed precipitation as a major driver of NPP. Temperature was negatively correlated with NPP in 99% of the regions at the annual scale, but after removing the effect of precipitation, temperature was positively correlated with the NPP in 77% of the regions.

## 1 Introduction

Studies of NPP and meteoretical factors provide the basis for theoretical and practical evaluation of local-to-global carbon cycles(Fang, 2002; Li et al., 2016b; Mowll et al., 2015). Derived relationships can also provide guidance for sustainable use of resources and realization of the productive potential of ecosystems(Gang et al., 2013; Li et al., 2013; Tan et al., 2007; Tang et al., 2010). Grasslands are important components of many terrestrial ecosystems(Wang et al., 2014a) and, due to their sensitivity to climate change, they are widely studied in that context(Reeves et al., 2014; Soylu et al., 2011; Wang et al., 2014b; Zhang et al., 2008). Process models are now widely used in exploring underlying biological processes in grasslands (such as photosynthesis and transpiration) as well as mechanisms driving interactions between these processes and environmental paramters(Liu et al., 2011).





Also in recent years, remote sensing data have been used in conjunction with models to estimate regional NPP(Chirici et al., 2015; Sjöström et al., 2013). For example, the Carnegie-Ames-Stanford Approach (CASA) has been used to estimate changes in the NPP of vegetation(Potter et al., 2012, 1993) as well as the efficiency at which NPP uses the products of photosynthetically active radiation (PAR)(Field et al., 1995). Typically, input data for CASA models used to predict NPP over large areas includes thematic mapper (TM) data(Yan et al., 2009) which provides surface distribution of NPP at a relatively high spatial resolution, as well as data from advanced very-high-resolution radiometer (AVHRR)(An et al., 2013), and moderate resolution imaging spectroradiometer (MODIS)(Li et al., 2016a; Sjöström et al., 2013).

Climate (temperature, precipitation, etc.) exerts critical control of vegetation growth in most ecosystems(Liu et al., 2013; Nemani et al., 2003; Reeves et al., 2014) and relationships between NPP and meteorological elements are widely studied using different methods(Michaletz et al., 2014). For example, linear regression and covariance have been used to assess relationships between aboveground NPP (ANPP) and temperature and precipitation (annually and during the growing season)(Mowll et al., 2015). For grassland ecosystems in Inner Mongolia, Zhang et al. estimated the spatial distribution of NPP in the Balager River Basin of the Xilingol Grassland using a light use efficiency model and analysed correlations among climate factors, vegetation indices and NPP. They found that precipitation and monthly mean temperature both correlated well with NPP and that precipitation had a greater impact than temperature(Zhang et al., 2015a). Mu et al. used remote sensing of the vegetation and the CASA model to reveal spatiotemporal dynamics of NPP for different types of vegetation as well as their differences in NPP responses to climate (Mu et al., 2013). Zhang et al. used the CENTURY model to simulate changes in the ANPP of grasslands in Xilinhot and their responses to climate change over the past 58 years. They showed that the ANPP of typical Inner Mongolian grasslands was highly sensitive to climate change, with distinct variation due to changes in temperature and precipitation(Zhang et al., 2012). Gao et al. found that NPP of semiarid grasslands of Inner Mongolia were significantly affected by biomass allocation and precipitation use efficiency(Gao et al., 2011). In addition, human management has a significant impact on NPP. Lkhagva et al. analysed the effect of grazing on community structure and ANPP of the semiarid grasslands and found that excessive grazing caused reductions in the distribution of bryophytic vegetation, the disappearance of frozen soils, and climate warming(Lkhagva et al., 2013).

Most previous studies have focused on NPP and its relationship to meteorological elements at either annual scales, or during the growing season. Here we attempt to investigate these relationships at different scales and the synergistic interactions between climatic variables. We assessed the NPP dynamics of a semiarid grassland (i.e., the Xilingol Grassland) between 2001 and 2013 using used a remote sensing data-based light use efficiency model in combination with other spatial and temporal data. Correlations and partial correlations between NPP and precipitation and temperature were analysed at the pixel level at both the annual and seasonal scales.





## 2 Materials and methods

### 2.1 General study area information

The Xilingol Grassland (115˚13'–117˚06'E and 43˚02'–44˚52'N) is located in the Xilingol League in the central Inner Mongolia Autonomous Region to the north of China (Fig.1). This grassland has a total area of 193,000 km$^2$ and a usable grassland area of 180,000 km$^2$. It can be divided into five main types: typical grasslands, desert grasslands, meadow grasslands, sandy grasslands and others. The study area has a northern temperate continental climate characterized by strong winds as well as arid conditions and cold temperatures. The mean annual temperature is 0–3˚C, and the multi-year mean precipitation is 295 mm. Precipitation gradually decreases from the southeast to the northwest and is mostly concentrated in July, August and September.

### 2.2 Data sources and processing

Remote sensing data used in this study - the 500-m×500-m, 8-day composite land surface reflectance product (MOD09A1) from 2001–2013 for the Xilingol League - were obtained from the Land Processes Distributed Active Archive Center of the United States Geological Survey (https://lpdaac.usgs.gov/). Normalized difference vegetation index (NDVI) data were obtained by calculation.

Meteorological data (including the monthly mean temperature (˚C), monthly precipitation (mm) and sunshine duration (h) from 2001–2013) were obtained from nine national standard meteorological stations of the China Meteorological Administration, namely, the East Ujimqin Banner, Erenhot, Naranbulag (Abag Banner), Abag Banner, Sonid Left Banner, Jurh (Sonid Right Banner), West Ujimqin Banner, Xilinhot and Duolun Meteorological Stations (Fig.1). Raster meteorological data are required for models of the vegetation NPP, and they were obtained by Kriging interpolation of data from the nine meteorological stations, using a module inverse distance weighted routine of the open geographic information system (GIS) software SAGA GIS version 2.2.7 (Conrad et al., 2015). Pixel size and projection type of the resulting raster data were consistent with the NDVI data.

We also employed data on livestock numbers in the nine counties in the Xilingol League between 2001 and 2013 (Statistical Yearbooks of the Xilingol League of 2002–2014)(Yang, 2015), including sheep and large livestock (cattle and horses). The method used by Li, 2007 was used to convert the data to the unit of a standard sheep (1 head of large livestock (cattle or horse) = 5 standard sheep) (Wen et al., 2007).

### 2.3 NPP estimation model

We applied the CASA model first developed by Potter et al. (1993) and Field et al. (1995) (Field et al., 1995; Potter et al., 1993), which is based on light use efficiency model. A modified version was developed by Zhu et al. (2007)(Zhu et al., 2007), and was used in this study. The main equations for estimating NPP are as follows:



$$NPP(x,t) = APAR(x,t) \times \varepsilon(x,t) \qquad (1)$$

$$APAR(x,t) = \frac{SOL(x,t) \times FPAR(x,t)}{2} \qquad (2)$$

where $SOL(x,t)$ represents the total solar radiation at pixel $x$ in month $t$ (MJ·m$^{-2}$·month$^{-1}$), and $FPAR(x,t)$ represents the fraction of the incident PAR absorbed by the vegetation.

The FPAR can be expressed based on the relationships between the PAR and a simple ratio index and NDVI. The method of calculation has been described in detail elsewhere(Field et al., 1995; Potter et al., 1993; Zhu et al., 2007).

**2.4 Method for verifying NPP estimation results**

Observation data are required for the verification of the NPP estimation results obtained by the CASA model, and the present study used the determination coefficient ($R^2$) and the root-mean-square error (RMSE) of the linear fit

(goodness-of-fit):

$$RMSE = \sqrt{\frac{\sum_{i=1}^{n}(P_i - O_i)^2}{n}} \qquad (3)$$

where $P_i$ and $O_i$ represent the estimated and observed values, respectively (i = 1, 2, ..., n, where n represents the number of samples).

**2.5 Correlation and partial correlation analyses between the NPP and the climate factors**

Pixel-based correlation coefficients and partial correlation coefficients between derived NPP and temperature and precipitation data were calculated at annual and seasonal scales to assess correlations between NPP and temperature and precipitation.

Correlation coefficients were calculated as follows:

$$R_{xy} = \frac{\sum_{i=1}^{n}\left[(x_i - \bar{x})(y_i - \bar{y})\right]}{\sqrt{\sum_{i=1}^{n}(x_i - \bar{x})^2 \sum_{i=1}^{n}(y_i - \bar{y})^2}} \qquad (4)$$

where $x$ and $y$ represent two variables; $\bar{x}$ and $\bar{y}$ represent the mean values of $x$ and $y$, respectively; $R_{xy}$ represents the correlation coefficient between $x$ and $y$; and $n$ represents the number of samples.

We used partial correlation analysis where:





$$r_{123} = \frac{r_{12} - r_{13}r_{23}}{\sqrt{\left(1 - r_{13}^2\right)\left(1 - r_{23}^2\right)}} \qquad (5)$$

and $r_{12}$, $r_{13}$ and $r_{23}$ represent the correlation coefficients between variables $X_1$ and $X_2$, between variables $X_1$ and $X_3$ and between variables $X_2$ and $X_3$, respectively. $r_{123}$ represents the partial correlation coefficient between $X_1$ and $X_2$ when $X_3$ is the control variable.

The partial correlation equation above was used to calculate partial correlation coefficients between NPP and temperature when precipitation was the control variable, as well as partial correlation coefficients between NPP and precipitation when temperature was the control variable.

## 3 Results

### 3.1 Verification of NPP estimates

Monitoring data from 46 monitoring stations within the Xilingol League collected in July 2011 (g C·m$^{-2}$·year$^{-1}$) were compared with simulated NPP for 2011 (Fig. 2). The correlation between simulated and measured values was based on geographic coordinates of each station (Fig. 1).

The correlation coefficient ($R^2$) of 0.67 indicates a reasonably strong, linear relationship between estimated and observed values. On this basis, we then used estimates from the CASA model to further analyse spatiotemporal

changes in NPP, as well as assess relationships to climate.

### 3.2 Inter-annual changes in the NPP, precipitation and temperature

Fig. 3a shows annual mean NPP, and NPP anomalies, for the vegetation in the Xilingol League between 2001 and 2013. Mean NPP varied between 141 and 313 g C·m$^{-2}$·year$^{-1}$, with a 13-year mean of 240 g C·m$^{-2}$·year$^{-1}$. Total NPP exhibited an increasing but insignificant ($r^2 = 0.11$, $p = 0.274$) trend with time.

Between 2001 and 2013, differences between annual NPP and long-term (13 year) means exhibited a sinusoidal shape. As with NPP, precipitation generally increased and was greatest (361 mm) in 2012, some 58% greater than the multi-year mean (Fig.3b). Precipitation totals for 2001 and 2005 were relatively low, some 29% and 28% less than multi-year mean precipitation, respectively. Mean annual temperatures varied between 4.58˚C in 2007 and 1.62˚C in 2012, with a period mean of 3.36˚C (Fig. 3c). Overall, mean annual temperature declined during the study period ($r^2 = 0.279$,

$p = 0.06$).

Fig. 4 shows the spatial distribution of NPP in the Xilingol League between 2001 and 2013. NPP in most regions was <500 g C·m$^{-2}$·year$^{-1}$. NPP in Abag Banner, Sonid Left Banner and Jurh in the western Xilingol League was between





100 and 300 g C·m$^{-2}$·year$^{-1}$, but was < 100 g C·m$^{-2}$·year$^{-1}$ in some regions of Erenhot. NPP significantly increased from west to east, and in West Ujimqin Banner and East Ujimqin Banner, NPP was between 300 and 700 g C·m$^{-2}$·year$^{-1}$.

### 3.3 Analysis of the relationships between NPP and precipitation and temperature

NPP, precipitation and temperature for each county were averaged according to season. Correlations between precipitation and temperature and NPP are shown in Table 1.

Annual NPP was mostly positively correlated with annual precipitation (Table 1). Strong correlations were observed for East Ujimqin Banner, Naranbulag (Abag Banner), Abag Banner, Jurh (Sonid Right Banner), Xilinhot and Duolun. Conversely, NPP was generally negatively correlated with mean temperature, albeit not significantly. These overall

patterns were not always borne out at regional and seasonal scales. For example, Spring temperatures were much more influential of Spring NPP than annual temperatures were of annual NPP. Similarly, Summer precipitation was particularly important to summer NPP (Table 1). There were numerous regional exceptions.  NPP for Erenhot, Sonid Left Banner and West Ujimqin Banner were seldom well predicted by either precipitation or temperature and only Spring temperatures had significant predictive power for NPP in these counties.

### 3.4 Spatial relationships between the annual NPP and precipitation and temperature

To further analyse spatial relationships between NPP and precipitation and temperature, we calculated correlation coefficients (R, Equation 4) and partial correlation coefficients (R$_p$, Equation 5) between the annual NPP of each pixel of the study area and annual precipitation and annual mean temperature between 2001 and 2013 (Fig. 5).

Using temperature as the control factor (Fig. 5a) there were no significant spatial or quantitative differences between R

and R$_p$ for the relationship between NPP and precipitation. In most regions in the Xilingol League, NPP was significantly positively correlated with precipitation, with R ranging from 0.6 to 1.0. NPP was negatively correlated with precipitation in only 0.32% of regions in the Xilingol League, with a mean correlation coefficient of 0.34. After the effect of the temperature was removed (R$_p$, Fig. 5b), there was almost no change in the relationship between the precipitation and the NPP in the study area.

Figs. 5c and d show negative correlations in most regions (99%) between NPP and temperature, before the removal of the precipitation effect. R ranged from -0.8 to 0. In around 1% of the regions in the study area, NPP was positively (but not significantly) correlated with annual mean temperature. After the effect of the precipitation was removed, 77% of regions showed a positive partial correlation between NPP and annual temperature (Fig. 5d).





## 3.5 Pixel-scale seasonal relationships between NPP and precipitation and temperature

Our study only investigated precipitation and temperature influences on NPP for Spring, Summer and Autumn owing to snow cover and lack of growth in Winter. At an even greater level of spatial detail, we calculated R and $R_p$ for relationships between NPP and climatic variables (precipitation and temperature) between 2001 and 2013 at the pixel scale. Fig. 6 a – f shows that R and $R_p$ for NPP and precipitation changed little across spring, summer and autumn. NPP was mostly positively correlated with precipitation. Negative relationships between NPP and precipitation were only significant in Spring, mostly after the effect of temperature was removed, and were largely confined to the south-west portion (Fig 6a, b).

Temperature effects on NPP were more variable (Fig. 6g – l). Mostly positive relationships in Spring were replaced by negative or neutral relationships in Summer (especially) and Autumn. High summer temperatures are clearly detrimental to NPP for much of the total study area. Most of the temperature effects were strongly mitigated by rainfall (contrast R and $R_p$ in summer).

## 4 Discussion

Our data for NPP in the study region are quantitatively similar to those reported using other approaches. For example, Li and Ji (2004)(Li and Ji, 2003) simulated NPP of grasslands throughout Inner Mongolia using the AVIMia model (Atmosphere-Vegetation Interaction Model and an impact assessment) and found that the multi-year mean NPP ranged from 223 to 315 g $C·m^{-2}·year^{-1}$. Zhu et al. (2007)(Zhu et al., 2007) assessed the distribution of NPP of the terrestrial vegetation of China between 1989 and 1993 using an improved CASA model with AVHRR NDVI values as the input data. Their results suggested that NPP of meadows, plain grasslands and desert grasslands was 383 g $C·m^{-2}·year^{-1}$, 226 g $C·m^{-2}·year^{-1}$ and 103 g $C·m^{-2}·year^{-1}$, respectively. Our data suggest NPP of the Xilingol Grassland slightly increased over the 13-year study period. Zhang (2014)(Zhang et al., 2014) analysed the dynamics of Xilingol Grassland in the growing season (April–October) between 2003 and 2012 using NDVI data and noted a similar trend. Li (2012)(Li et al., 2012) also used a NDVI-based method and recorded that the condition of the grasslands had improved between 2000 and 2006, consistent with results obtained by Jiang et al. (2006) through experimentation at fixed locations(Jiang et al., 2006). The analysis provided here, via integration of validated modelling with remote sensing, offers opportunity to extend such studies to other grassland regions in China and more globally.

Precipitation and temperature are well-known climatic influences on grassland productivity (Su et al., 2015; Wu et al., 2011), with the former being especially significant in arid regions(Li, 2000; Lü et al., 2014; Yang et al., 2008). The results of this study support this general interpretation; positive correlations between annual NPP and precipitation in most regions of the study area demonstrate the strength of control. In addition, negative correlations between





temperature and NPP were mostly conditional upon precipitation. These effects are seen most clearly via the differences between R and $R_p$ for the relationships of precipitation and temperature to NPP (Fig. 5a and b and Fig. 6a–f). More detailed analysis (see Fig. A1a, c, e and g) shows that the distribution ranges of correlation coefficients for the relationship between NPP and precipitation, were unaffected by removing temperature. In simple terms, temperature

had no impact on the relationship between precipitation and NPP. Conversely, temperature effects were clearly precipitation dependent. For example at the annual scale, when precipitation effects were included, NPP in the study area was negatively correlated with temperature in 99% of regions (Fig. 5c and Fig. A1b). However when precipitation was removed (Fig. 5d and Fig. A1b), there was a positive correlation in 77% of regions. This pattern is clearly related to the biology of plant growth. Increased temperatures when water is readily available stimulates growth(Dou et al.,

2009; Huang et al., 2014). Under drought conditions, high temperatures can severely reduce growth(Shen et al., 2012). Consequently, correlations and partial correlations between NPP and temperature were consistently positive in Spring (Fig. 6g and h and Fig. A1d) and mainly negative in Summer (Fig. 6i and j and Fig. A1f) and autumn (Fig. 6k and l and Fig. A1h).

As is commonly recognized, aside from precipitation and temperature, NPP is also subject to the influence of other

environmental/climate factors and human activities(Luo et al., 2008; Zhang et al., 2015b). For example, while Zhao (2010)(Zhao and Running, 2010) found that high temperatures and droughts between 2000 and 2009 were primary causes of reduced global NPP, and Han et al. (2006) found that precipitation and temperature, contributed almost 60% of the variation in the total biomass(Han et al., 2006), human activities remain one of the main reported causes of grassland degradation(Akiyama and Kawamura, 2007; Zhao et al., 2005; Zheng et al., 2006). Our assessment is that

numbers of grazing livestock declined between 2001 and 2013, and were not significantly related to NPP (Fig. A2). In contrast, using a NDVI-based method Li et al. (2012) concluded that human activities (grazing) were the main driving factor of changes in the vegetation between 1981 and 2006(Li et al., 2012).

## 5 Conclusions

Through the calculation of NPP of the Xilingol Grassland between 2001 and 2013, the present study analysed the

relationships between the NPP and the climate factors at different time scales and then performed a comparison analysis to determine the effect of the climate factors on NPP of the vegetation. The results show that the CASA model can be used to estimate NPP of the Xilingol Grassland. These estimates show that the inter-annual change in NPP between 2001 and 2013 exhibited a slight increasing, albeit insignificant, trend over the 13-year period. The correlation and partial correlation analyses of the NPP and the precipitation and temperature show that the NPP was

relatively highly positively correlated with the precipitation regardless of the temporal scale, but its relationship with temperature varied between the annual and seasonal time scales. At the annual scale, the correlation coefficients



between the NPP and the temperature were mostly negative, whereas the partial correlations were mostly positive. The relationship between NPP and temperature also varied between different seasons, and due to the relatively low precipitation, the NPP was mainly affected by the temperature in spring. However, the NPP was controlled by both the precipitation and temperature in summer and autumn. In addition, a simple analysis of the relationship between grazing

and the NPP was also performed in the study period, and the NPP was not found to be significantly correlated with grazing intensity.

**Acknowledgements**

This research was supported by the National Natural Science Foundation of China (No. 51569017, 51269014, 51139002, 31360113, and 51169013), the Program for Changjiang Scholars and Innovative Research Team of Chinese Ministry of Education (IRT13069), the Natural Science Foundation of Inner Mongolia (2015MS0514) and the Western Region China Postdoctoral Science Foundation (2015M572630XB). Further, we would like to thank the two anonymous reviewers for substantially improving the manuscript.

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





| Station Names of stations | Correlation coefficients between the NPP and the climate factors | | | | | | | |
| | Annual NPP | | Spring NPP | | Summer NPP | | Autumn NPP | |
| | AP | AMT | SpP | SpMT | SuP | SuMT | AuP | AuMT |
| East Ujimqin Banner | 0.594* | -0.126 | 0.632* | 0.560* | 0.531 | -0.3 | 0.826** | 0.073 |
| Erenhot | 0.135 | 0.131 | 0.303 | 0.777** | 0.281 | -0.336 | 0.558* | 0.137 |
| Naranbulag (Abag Banner) | 0.828** | -0.452 | 0.618* | 0.654* | 0.803** | -0.174 | 0.732** | -0.079 |
| Abag Banner | 0.775** | -0.383 | 0.657* | 0.610* | 0.699** | -0.416 | 0.761** | -0.287 |
| Sonid Left Banner | -0.042 | 0.402 | 0.54 | 0.808** | 0.094 | -0.255 | 0.337 | 0.326 |
| Jurh (Sonid Right Banner) | 0.690** | -0.126 | 0.582* | 0.759** | 0.665* | -0.054 | 0.605* | 0.079 |
| West Ujimqin Banner | 0.465 | -0.107 | 0.382 | 0.762** | 0.534 | -0.251 | 0.275 | 0.267 |
| Xilinhot | 0.682* | -0.304 | 0.453 | 0.751** | 0.53 | -0.525 | 0.738** | -0.089 |
| Duolun | 0.778** | -0.284 | 0.151 | 0.701** | 0.646* | -0.467 | 0.748** | -0.331 |

**Table 1. Correlation coefficients (r) between annual and seasonal NPP and climate variables**

5  **AP = annual mean precipitation; AMT = annual mean temperature; SpP = spring precipitation; SpMT = spring mean temperature; SuP = summer precipitation; SuMT = summer mean temperature; AuP = autumn precipitation; AuMT = autumn mean temperature.**
**\* indicates a significant correlation at the 0.05 level (two-tailed).**
**\*\* indicates a significant correlation at the 0.01 level (two-tailed).**




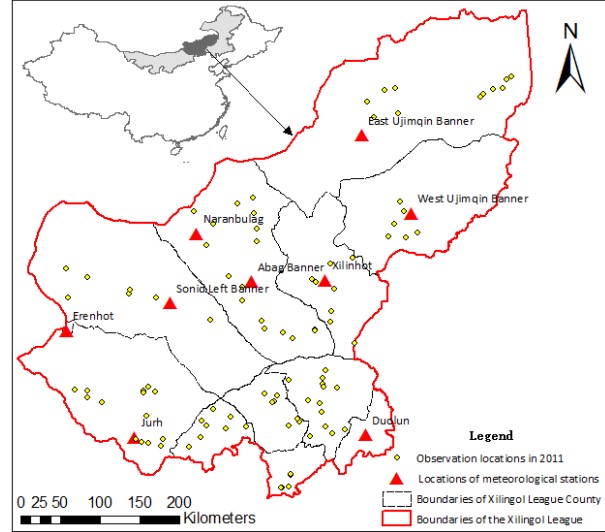

**Figure 1. Study area and locations of meteorological stations and observation locations in 2011.**





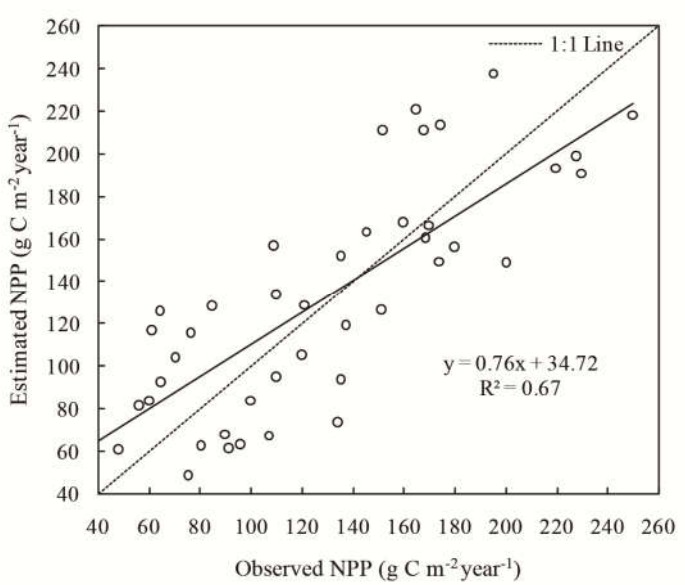

**Figure 2. Correlation between the estimated and observed NPP**





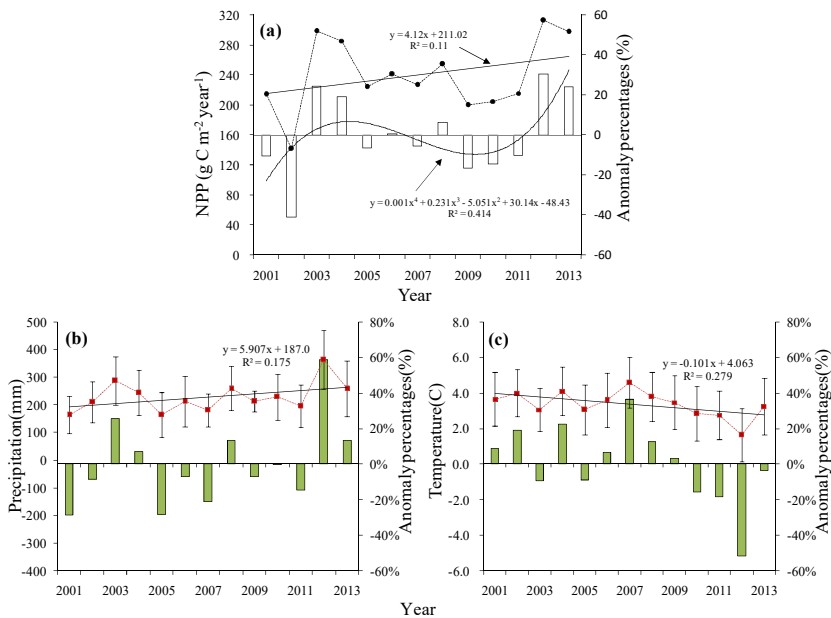

**Figure 3. Changes in the NPP, precipitation and temperature in the study area between 2001 and 2013.**

**a: Mean NPP (dotted line), linear regression of the mean NPP (straight line), difference between annual NPP and long-term NPP**

**(histogram), and linear regression of the difference between annual NPP and long-term NPP (straight line). b: Mean precipitation**

**(dotted line), linear regression of the mean precipitation (straight line) and difference between annual and long-term precipitation**

**(histogram). c: Mean temperature (dotted line), linear regression of mean temperature and difference between annual and long-**

**term temperature (histogram).**





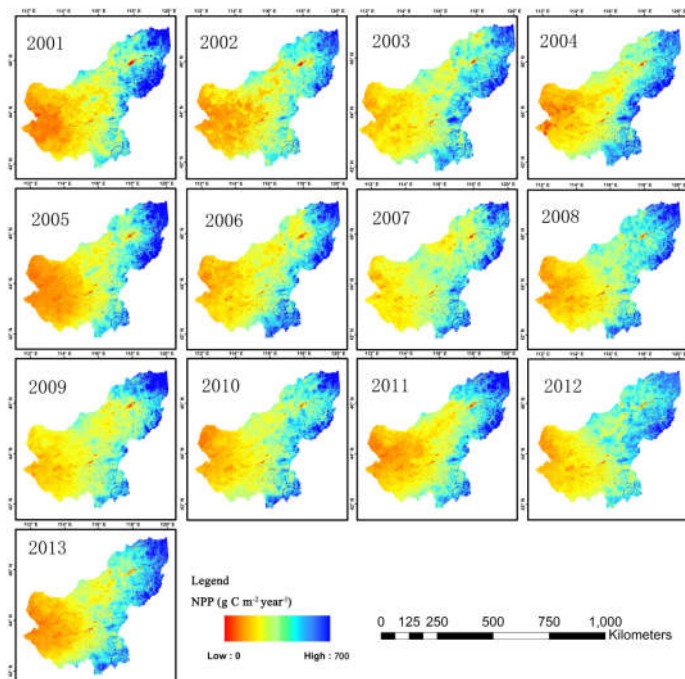

**Figure 4. Spatial distribution of the annual total NPP between 2001 and 2013**




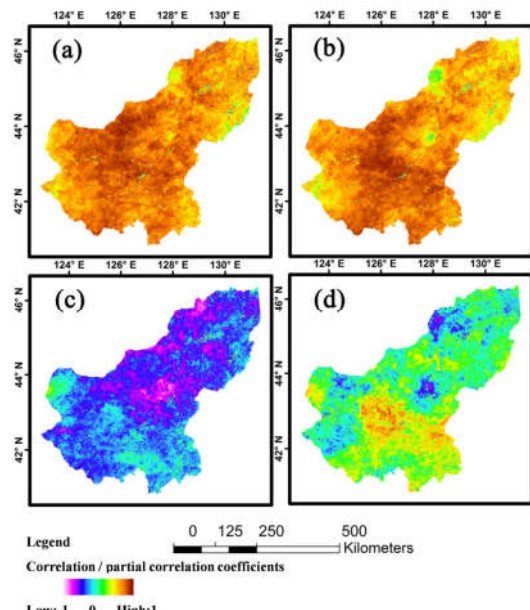

**Figure 5. Correlation coefficients and partial correlation coefficients between annual mean NPP and climate variables**

a: Correlation analysis between NPP and precipitation. b: Partial correlation analysis between NPP and precipitation. c: Correlation analysis between NPP and temperature. d: Partial correlation analysis between NPP and temperature.




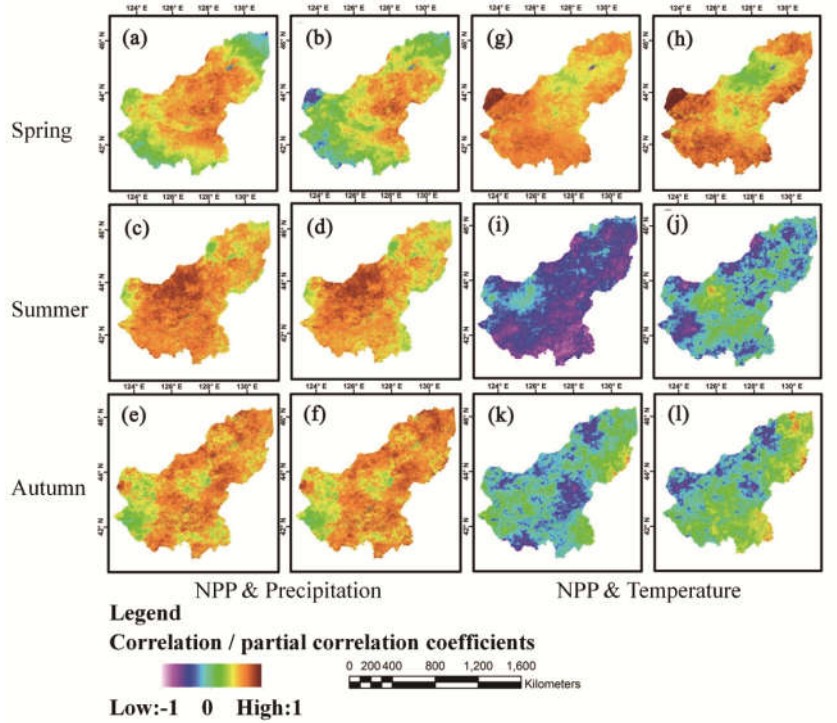

**Figure 6. Analysis of correlations and partial correlations between NPP and climate variables across seasons.**

a, c, e Correlation analysis of NPP and precipitation. b, d, f Partial correlation analysis of NPP and precipitation. g, i, k Correlation analysis of NPP and temperature. h, j, l Partial correlation analysis of NPP and temperature.





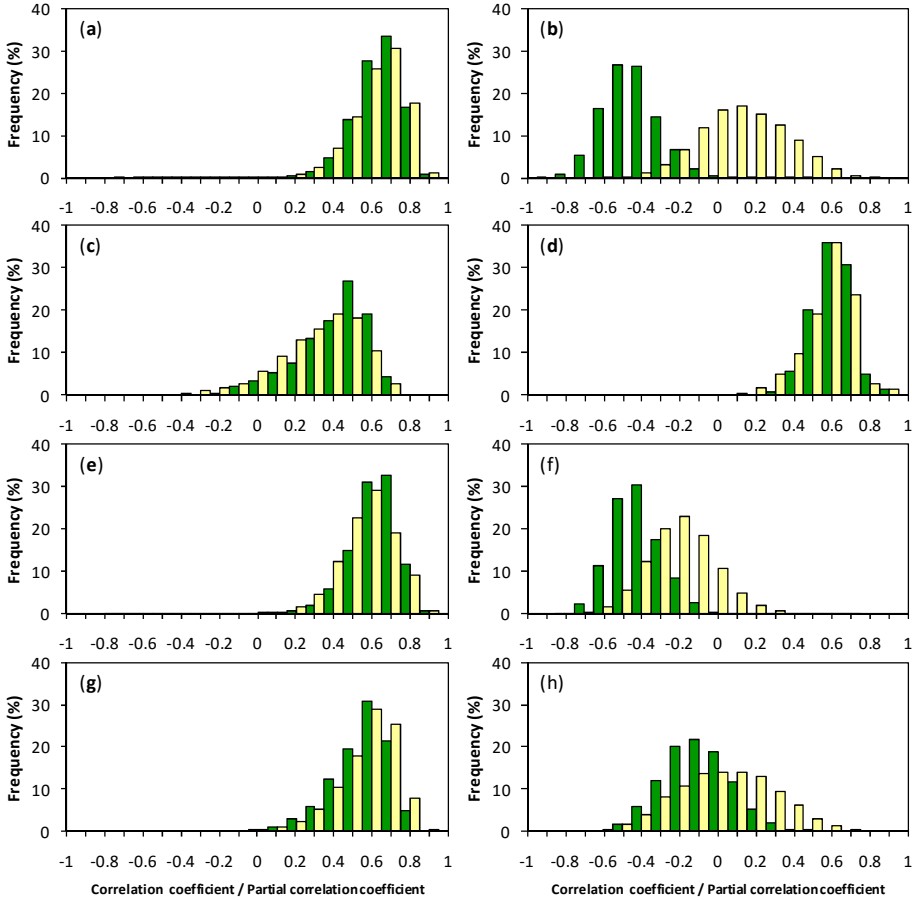

**Figure A1. Histograms of the correlation coefficients (dark green) and partial correlation coefficients (light yellow) between the**

**NPP and the precipitation and temperature.**

a: Annual NPP and precipitation. b: Annual NPP and temperature. c: NPP and precipitation in spring. d: NPP and temperature
in spring. e: NPP and precipitation in summer. f: NPP and temperature in summer. g: NPP and precipitation in autumn. h: NPP
and temperature in autumn.





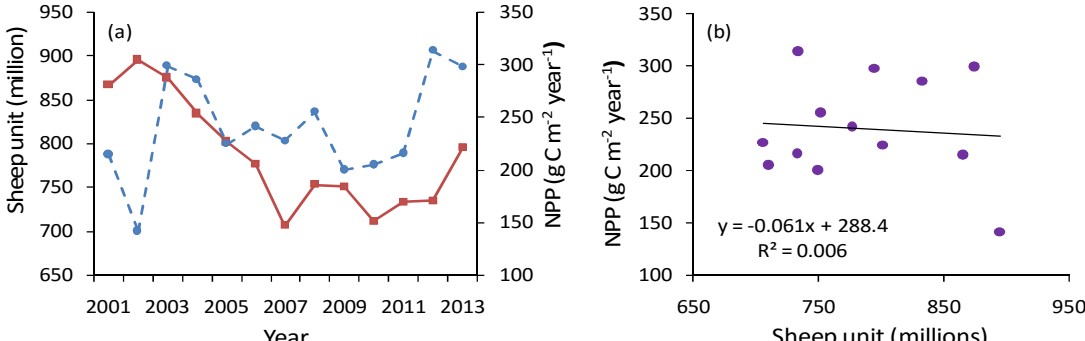

**Figure A2. (a) Grazing intensity (solid red line) and change in the NPP (broken blue line) and (b) the correlation between the intensity of grazing and the change in the NPP**