# Peer review of "Empirical and model-based estimates of spatial and temporal variations in net primary productivity in semi-arid grasslands of Northern China"

_Hydrology and Earth System Sciences, 2016_

## Referee Comment (RC1) · Anonymous Referee #1 · 12 Feb 2017

The authors used a light use efficient model (LUE model, the CASA model) to estimate the net primary productivity (NPP) in grasslands over northern China from 2001-2013. Then they examined the precipitation and temperature influences on the modeled NPP in different seasons. While this study has been carried out with great efforts, some issues have remained in the paper as described below.

1. There are quite a few studies on the NPP distribution over China at regional and national scales, using LUE models or process-based models. However, the literature review on these previous studies is not thorough. It seems that this study has not

[Figure]

made much advance from the earlier studies. The authors used the CASA model for a regional NPP estimation for a number of years. This is not scientifically and technically novel and challenging. Such a regional application may not have broad implications so the authors may submit their paper to a journal more for regional applications.

2. The discussion on the precipitation and temperature influences are reasonable but these influences are commonly known and are available in literatures.

3. There are many critical descriptions missing in the paper that hampers the reader from understanding how NPP is derived. For example, in Equation 1, how is $\varepsilon$ derived? Does it depend on vegetation type? If so, how? Scientists have improved the expression of $\varepsilon$ since the LUE idea was first proposed and thus $\varepsilon$ can depend on several environmental variables. Is this also the case for this study? If so, how? How is soil moisture considered in the model? Or has the model not considered soil moisture in this application? The authors only provided the source for NDVI in the model. How about SOL and FPAR?

---

## Referee Comment (RC2) · Anonymous Referee #2 · 17 Feb 2017

The paper entitled "Empirical and model-based estimates of spatial and temporal variations in net primary productivity in semi-arid grasslands of Northern China" by S. Zhang et al. presents an analysis on the impact of both temperature and precipitation on grassland NPP estimations obtained using a light-use efficiency model. The topic (climate control on NPP), although interesting, is far from the HESS scope and is probably more suitable for the sister EGU journal Biogeosciences. The paper is methodologically obscure as both the modelling approach and datasets applied in the study are poorly documented, which strongly hinders the reproducibility of the study. Furthermore, I have major concerns on the type of model chosen for estimating NPP, the data

quality, and the (too simplistic) analysis presented in the study about the climate-driven controls on NPP.

Major concerns:

1. Model type: The authors applied the CASA model, a light-use efficiency model, to simulate the inter-annual dynamics of semi-arid grassland NPP. Vegetation production in drylands is limited mainly by water availability. Any attempt to model the dynamics of vegetation production in semi-arid landscapes must consider, at least, the dynamics of soil moisture availability.

2. Model details: The authors apparently fed the CASA model using MODIS NDVI data for estimating NPP. The model is described in the study in just two equations: an equation for APAR (that is proportional to FPAR and solar radiation) and another equation linking NPP with APAR. No details are described on how FPAR and solar radiation have been calculated. Furthermore, it is not clear how the authors have applied the NDVI data to feed the model. In fact, the variable NDVI is not included in the equations.

3. Use of NDVI data: In arid and semi-arid landscapes, where vegetation cover is sparse and generally low, NDVI data is strongly affected by the soil background properties. Bearing in mind that the study covers a very broad area of approx. 200,000 Km2 where soil characteristics can change dramatically between locations, the use of NDVI data is undesirable. The use of either EVI or MSAVI is probably far more appropriate for this application.

4. Verification of NPP estimates: The authors indicate that "monitoring data from 46 monitoring stations within the Xilingol League collected in July 2011 (g C m-2 year-1) were compared with simulated NPP for 2011". However, the field-based NPP data is not described in any way in the paper. How was NPP measured in the field? What size were the plots? Were the NPP estimations obtained by a single harvest in July 2011? Please, note that a single harvest of aboveground biomass does not represent

accurately NPP in perennial grasslands (the type of vegetation analyzed in the paper). Biomass harvests must be taken both at the beginning and at the peak of the growing season to obtain a valid NPP estimate.

5. Climate data: Meteorological data for both the NPP estimations and the analysis of the NPP-climate relations was obtained by simple kriging interpolation from nine meteorological stations. Nine stations for an area of 193,000 km2 is probably too little information to sustain an accurate estimation of spatially-distributed climate/meteorological variables for the full area. Furthermore, I expect that, in an area as big as 193,000 km2, topographical variations (e.g. local differences in elevation) can have a very important role in local climate and meteorology. Did the authors check for the influence of elevation and other topographical variables on the meteorological records of the stations? The use of kriging with varying local means, kriging with external drift and co-kriging can improve considerably the spatial interpolation of meteorological variables where local elevations (and other topographical factors) have a relevant role.

6. Livestock stocking density data: The authors indicate the source for the livestock data (a paper by Yang, 2015), but should also detail how this data was generated, since the source is in Chinese. Furthermore, the data is expressed in animal units (e.g. in Fig. A2) and should be expressed in density units (i.e. animals km-2).

7. Data analysis: The authors apply simple correlations to analyze the impact of temperature and rainfall on NPP. This type of analysis is too simplistic and does not provide any novel information to that already published on this topic.

---

## Author Comment (AC1) · 14 Mar 2017

**Reply to Anonymous Referee #1**

We thank Referee #1 for the comments; below we give the reply to the comments.

The authors used a light use efficient model (LUE model, the CASA model) to estimate the net primary productivity (NPP) in grasslands over northern China from 2001-2013. Then they examined the precipitation and temperature influences on the modeled NPP in different seasons. While this study has been carried out with great efforts, some issues have remained in the paper as described below.
1. There are quite a few studies on the NPP distribution over China at regional and national scales, using LUE models or process-based models. However, the literature review on these previous studies is not thorough. It seems that this study has not made much advance from the earlier studies. The authors used the CASA model for a regional NPP estimation for a number of years. This is not scientifically and technically novel and challenging. Such a regional application may not have broad implications so the authors may submit their paper to a journal more for regional applications.

**Reply:** thanks for the suggestions. We added some literatures in introductions part as showed in the revised version of the paper.

2. The discussion on the precipitation and temperature influences are reasonable but these influences are commonly known and are available in literatures.

**Reply:** thanks and we agree to the reviewer's comment. But in this paper we gave the quantitative different about correlations and partial correlations between NPP and temperature and precipitation, also we analysis the relationship between grazing and NPP in our study area it is different with other research results.

3. There are many critical descriptions missing in the paper that hampers the reader from understanding how NPP is derived. For example, in Equation 1, how is ε derived? Does it depend on vegetation type? If so, how? Scientists have improved the expression of ε since the LUE idea was first proposed and thus ε can depend on several environmental variables. Is this also the case for this study? If so, how? How is soil moisture considered in the model? Or has the model not considered soil moisture in this application? The authors only provided the source for NDVI in the model. How about SOL and FPAR?

**Reply:** thanks and we agree to the reviewer's comment, we add some equations and describe about how the NPP derived and more describe about the input data. Which showed in the revised version of the paper.

**Reply to Anonymous Referee #2**

We thank Referee #1 for the comments; below we give the reply to the comments.

The paper entitled "Empirical and model-based estimates of spatial and temporal variations in net primary productivity in semi-arid grasslands of Northern China" by S. Zhang et al. presents an analysis on the impact of both temperature and precipitation on grassland NPP estimations obtained using a light-use efficiency model. The topic (climate control on NPP), although interesting, is far from the HESS scope and is probably more suitable for the sister EGU journal Biogeosciences. The paper is methodologically obscure as both the modelling approach and datasets applied in the study are poorly documented, which strongly hinders the reproducibility of the study. Furthermore, I have major concerns on the type of model chosen for estimating NPP, the data quality, and the (too simplistic) analysis presented in the study about the climate-driven controls on NPP.

Major concerns:

1. Model type: The authors applied the CASA model, a light-use efficiency model, to simulate the inter-annual dynamics of semi-arid grassland NPP. Vegetation production in drylands is limited mainly by water availability. Any attempt to model the dynamics of vegetation production in semi-arid landscapes must consider, at least, the dynamics of soil moisture availability.

**Reply:** thanks and we agree to the reviewer's comment. In CASA model the water stress is defined as moisture stress coefficient ($W_\varepsilon(x, t)$), we added the describe about it in the revised version of the paper.

2. Model details: The authors apparently fed the CASA model using MODIS NDVI data for estimating NPP. The model is described in the study in just two equations: an equation for APAR (that is proportional to FPAR and solar radiation) and another equation linking NPP with APAR. No details are described on how FPAR and solar radiation have been calculated. Furthermore, it is not clear how the authors have applied the NDVI data to feed the model. In fact, the variable NDVI is not included in the equations.

**Reply:** thanks and we agree to the reviewer's comment. We added some equations and describe about how the NPP derived and more describe about the input data. Which showed in the revised version of the paper.

3. Use of NDVI data: In arid and semi-arid landscapes, where vegetation cover is sparse and generally low, NDVI data is strongly affected by the soil background properties. Bearing in mind that the study covers a very broad area of approx. 200,000 Km2 where soil characteristics can change dramatically between locations, the use of NDVI data is undesirable. The use of either EVI or MSAVI is probably far more appropriate for this

application.

**Reply:** thanks for the comment, the CASA model uses NDVI data to estimated NPP had been test a lots from globe to regional scale with variations remote sensing data as mentioned in the introduction of the revised version of this paper.

4. Verification of NPP estimates: The authors indicate that "monitoring data from 46 monitoring stations within the Xilingol League collected in July 2011 (g C m-2 year-1) were compared with simulated NPP for 2011". However, the field-based NPP data is not described in any way in the paper. How was NPP measured in the field? What size were the plots? Were the NPP estimations obtained by a single harvest in July 2011? Please, note that a single harvest of aboveground biomass does not represent accurately NPP in perennial grasslands (the type of vegetation analyzed in the paper). Biomass harvests must be taken both at the beginning and at the peak of the growing season to obtain a valid NPP estimate.

**Reply:** thanks and we agree to the reviewer's comment. More detail was added about the filed NPP in the revised version of this paper.

5. Climate data: Meteorological data for both the NPP estimations and the analysis of the NPP-climate relations was obtained by simple kriging interpolation from nine meteorological stations. Nine stations for an area of 193,000 km2 is probably too little information to sustain an accurate estimation of spatially-distributed climate/meteorological variables for the full area. Furthermore, I expect that, in an area as big as 193,000 km2, topographical variations (e.g. local differences in elevation) can have a very important role in local climate and meteorology. Did the authors check for the influence of elevation and other topographical variables on the meteorological records of the stations? The use of kriging with varying local means, kriging with external drift and co-kriging can improve considerably the spatial interpolation of meteorological variables where local elevations (and other topographical factors) have a relevant role.

**Reply:** thanks for the comment. We didn't use co-kriging to interpolate meteorological data. Because in mostly of the papers used CASA to simulate NPP they use kriging or IDW to interpolate climate data, we did this according to those literatures.

6. Livestock stocking density data: The authors indicate the source for the livestock data (a paper by Yang, 2015), but should also detail how this data was generated, since the source is in Chinese. Furthermore, the data is expressed in animal units (e.g. in Fig. A2) and should be expressed in density units (i.e. animals km-2).

**Reply:** thanks and we agree to the reviewer's comment. More detail was added about the livestock data in the revised version of the paper. The data in Fig. A2 can be showed in density or in total number because the area is same in different year. But we changed

the title to "Figure A2. (a) Number of livestock (solid red line) and change in the NPP (broken blue line) and (b) the correlation between the number of livestock and the change in the NPP" to make clear for the reader.

7. Data analysis: The authors apply simple correlations to analyze the impact of temperature and rainfall on NPP. This type of analysis is too simplistic and does not provide any novel information to that already published on this topic.

**Reply:** thanks for the comment. Same answer as reply to the comment 2 of Referee #1, in this paper we gave the quantitative different about correlations and partial correlations between NPP and temperature and precipitation, also we analysis the relationship between grazing and NPP in our study area it is different with other research results.

**Revised version of the paper.**

[revised manuscript text omitted]

Observed NPP were collected from 46 monitoring stations within the Xilingol in July of 2011 as Fig 1 shows. Three pairs of 0.5m×0.5m quadrats sample plots were investigeated in each monitoing stations. Plant's species, high and coverage were investigated, and then all vegetation was clipped at the soil surface and dried in lab at 75°C for 48 h prior to weighing. Aboveground biomass (AGB) was estimated by averaging the biomass of three plots. Belowground biomass (BGB) was collected by root augers with 8.9 cm diameter corresponding to AGB and dried as AGB. Then the total biomass was the sum of AGB and BGB. The conversion coefficient 0.475 was used to conversion biomass(g·m$^{-2}$) to NPP(gC·m$^{-2}$·a$^{-1}$) (Raich et al., 1991).

We also employed data on livestock numbers in the nine counties in the Xilingol League between 2001 and 2013 from the Statistical Yearbooks of the Xilingol League of 2002–2014 (Yang, 2015), including sheep and large livestock (cattle and horses). According to the rules of National Bureau of Statistics of the People's Republic of China since 2008 all the numbers of horse, cow and sheep are from sampling survey. The method used by Li, 2007 was used to convert the data to the unit of a standard sheep (1 head of large livestock (cattle or horse) = 5 standard sheep) (Wen et al., 2007).

**2.3 NPP estimation model**

We applied the CASA model first developed by Potter et al. (1993) and Field et al. (1995) (Field et al., 1995; Potter et al., 1993), which is based on light use efficiency model. A modified version was

developed by Zhu et al(Zhu et al., 2007), and was used in this study. The main equations for estimating NPP are as follows:

$$NPP(x,t) = APAR(x,t) \times \varepsilon(x,t) \qquad (1)$$

$$APAR(x,t) = SOL(x,t) \times FPAR(x,t) \times 0.5 \qquad (2)$$

where *SOL(x,t)* represents the total solar radiation at pixel *x* in month *t* (MJ m$^{-2}$ month$^{-1}$), and *FPAR(x,t)* represents the fraction of the incident PAR absorbed by the vegetation, The value of 0.5 stands for the fraction of total solar radiation that can be used by vegetation (0.38–0.71mm). The FPAR can be expressed based on the relationships between FPAR and NDVI as well as Simple Ratio (SR), which are calculated from Eqs 3 to 6:

$$FPAR(x,t) = [FPAR(x,t)_{SR} + FPAR(x,t)_{NDVI}]/2 \qquad (3)$$

$$FPAR(x,t)_{NDVI} = \frac{(NDVI(x,t) - NDVI_{min})}{(NDVI_{i,max} - NDVI_{i,min})} \times (FPAR_{max} - FPAR_{min}) + FPAR_{min} \qquad (4)$$

$$FPAR(x,t)_{SR} = \frac{(SR(x,t) - SR_{min})}{(SR_{max} - SR_{min})} \times (FPAR_{max} - FPAR_{min}) + FPAR_{min} \qquad (5)$$

$$SR(x,t) = [1 + NDVI(x,t)]/[1 - NDVI(x,t)] \qquad (6)$$

Where *NDVI$_{i,max}$* and *NDVI$_{i,min}$* is the maxim and minim value of NDVI corresponding to different plant types obtained from Land Cover Products of China, which is provided by Environmental and Ecological Science Data Center for West China, National Natural Science Foundation of China (http://westdc.westgis.ac.cn)(Youhua et al., 2010),    *FPAR$_{min}$* and *FPAR$_{max}$* is 0.001 and 0.95 which is independent with vegetation types. *SR$_{i,max}$* and *SR$_i$,min* represent the 95% and 5% of NDVI of the different vegetation.

The algorithm for light use efficiency can be expressed as follows:

$$\varepsilon(x,t) = T_{\varepsilon 1}(x,t) \times T_{\varepsilon 2}(x,t) \times W_{\varepsilon}(x,t) \times \varepsilon_{max} \qquad (7)$$

where $T_{\varepsilon 1}(x,t)$ and $T_{\varepsilon 2}(x,t)$ are the temperature stress coefficients, which reflect the reduction of light-use efficiency caused by a temperature factor (Field et al., 1995), $W_{\varepsilon}(x,t)$ is the moisture stress coefficient which indicates the reduction in light use efficiency caused by the moisture factor(Field et al., 1995), and $\varepsilon_{max}$ is the maximum light use efficiency under ideal conditions and can be set to different constant parameters for different vegetation types. The value of $\varepsilon_{max}$ for grassland is 0.542 gC·MJ$^{-1}$ and 0.429 gC MJ$^{-1}$ for shrubs in this study, in accordance with the study of (Zhu et al., 2007). A more detailed description of this algorithm can be found in(Yu et al., 2011; Zhu et al., 2007).

**2.4 Method for verifying NPP estimation results**

Observation data are required for the verification of the NPP estimation results obtained by the CASA model, and the present study used the determination coefficient (R$^2$) and the root-mean-square error (RMSE) of the linear fit (goodness-of-fit):

$$RMSE = \sqrt{\frac{\sum_{i=1}^{n}(P_i - O_i)^2}{n}} \qquad (8)$$

where *P$_i$* and *O$_i$* represent the estimated and observed values, respectively (i = 1, 2, …, n, where n represents the number of samples).

**2.5 Correlation and partial correlation analyses between the NPP and the climate factors**

Pixel-based correlation coefficients and partial correlation coefficients between derived NPP and temperature and precipitation data were calculated at annual and seasonal scales to assess correlations between NPP and temperature and precipitation.
Correlation coefficients were calculated as follows:

$$R_{xy} = \frac{\sum_{i=1}^{n} \left[ (x_i - \bar{x})(y_i - \bar{y}) \right]}{\sqrt{\sum_{i=1}^{n}(x_i - \bar{x})^2 \sum_{i=1}^{n}(y_i - \bar{y})^2}} \tag{9}$$

where $x$ and $y$ represent two variables; $\bar{x}$ and $\bar{y}$ represent the mean values of $x$ and $y$, respectively; $R_{xy}$ represents the correlation coefficient between $x$ and $y$; and $n$ represents the number of samples.
We used partial correlation analysis where:

$$r_{123} = \frac{r_{12} - r_{13}r_{23}}{\sqrt{\left(1 - r_{13}^2\right)\left(1 - r_{23}^2\right)}} \tag{10}$$

and $r_{12}$, $r_{13}$ and $r_{23}$ represent the correlation coefficients between variables $X_1$ and $X_2$, between variables $X_1$ and $X_3$ and between variables $X_2$ and $X_3$, respectively. $r_{123}$ represents the partial correlation coefficient between $X_1$ and $X_2$ when $X_3$ is the control variable.
The partial correlation equation above was used to calculate partial correlation coefficients between NPP and temperature when precipitation was the control variable, as well as partial correlation coefficients between NPP and precipitation when temperature was the control variable.

**3 Results**

**3.1 Verification of NPP estimates**

Monitoring data from 46 monitoring stations were compared with simulated NPP for 2011 (Fig. 2). The estimated NPP was the sum of NPP from January to July because the investigating data was collected from the end of July. 
[revised manuscript text omitted]

Raich, A. J. W., Rastetter, E. B., Melillo, J. M., Kicklighter, D. W., Steudler, P. A., Grace, A. L., Iii, B.

M. and Vörösmarty, C. J.: Potential Net Primary Productivity in South America : Application of a

Global Model, Ecological Applications, 1(4), 399–429 [online] Available from:

http://www.jstor.org/stable/1941899 (Accessed 22 February 2017), 1991.

Reeves, M. C., Moreno, A. L., Bagne, K. E. and Running, S. W.: Estimating climate change effects

on net primary production of rangelands in the United States, Climatic Change, 126(3–4), 429–

442, doi:10.1007/s10584-014-1235-8, 2014.

Shen, W., Li, H., Sun, M. and Jiang, J.: Dynamics of aeolian sandy land in the Yarlung Zangbo River

basin of Tibet, China from 1975 to 2008, Global and Planetary Change, 86, 37–44,

doi:10.1016/j.gloplacha.2012.01.012, 2012.

Sjöström, M., Zhao, M., Archibald, S., Arneth, A., Cappelaere, B., Falk, U., de Grandcourt, A.,

Hanan, N., Kergoat, L., Kutsch, W., Merbold, L., Mougin, E., Nickless, A., Nouvellon, Y., Scholes, R.

J. J., Veenendaal, E. M. M. and Ardö, J.: Evaluation of MODIS gross primary productivity for Africa

using eddy covariance data, Remote Sensing of Environment, 131, 275–286,

doi:10.1016/j.rse.2012.12.023, 2013.

Soylu, M. E., Istanbulluoglu, E., Lenters, J. D. and Wang, T.: Quantifying the impact of

groundwater depth on evapotranspiration in a semi-arid grassland region, Hydrology and Earth

System Sciences, 15(3), 787–806, doi:10.5194/hess-15-787-2011, 2011.

Su, H., Feng, J., Axmacher, J. C. and Sang, W.: Asymmetric warming significantly affects net

primary production, but not ecosystem carbon balances of forest and grassland ecosystems in

northern China, Scientific Reports, 5, 9115, doi:10.1038/srep09115, 2015.

Tan, K., Piao, S., Peng, C. and Fang, J.: Satellite-based estimation of biomass carbon stocks for

northeast China's forests between 1982 and 1999, Forest Ecology and Management, 240(1), 114–

121, doi:10.1016/j.foreco.2006.12.018, 2007.

Tang, G., Beckage, B., Smith, B. and Miller, P. A.: Estimating potential forest NPP, biomass and

their climatic sensitivity in New England using a dynamic ecosystem model, Ecosphere, 1(6),

art18, doi:10.1890/ES10-00087.1, 2010.

Wang, C., Han, G., Wang, S., Zhai, X., Brown, J., Havstad, K. M., Ma, X., Wilkes, A., Zhao, M., Tang,

S., Zhou, P., Jiang, Y., Lu, T., Wang, Z. and Li, Z.: Sound management may sequester methane in

grazed rangeland ecosystems, Scientific Reports, 4, 157–158, doi:10.1038/srep04444, 2014a.

Wang, X., Li, F., Gao, R., Luo, Y. and Liu, T.: Predicted NPP spatiotemporal variations in a semiarid

steppe watershed for historical and trending climates, Journal of Arid Environments, 104, 67–79,

doi:10.1016/j.jaridenv.2014.02.003, 2014b.

Wang, Y. and Wesche, K.: Vegetation and soil responses to livestock grazing in Central Asian

grasslands: a review of Chinese literature, Biodiversity and Conservation, 25(12), 2401–2420,

doi:10.1007/s10531-015-1034-1, 2016.

Wen, J., Ali, S. H. and Zhang, Q.: Property rights and grassland degradation: A study of the Xilingol

Pasture, Inner Mongolia, China, Journal of Environmental Management, 85(2), 461–470,

doi:10.1016/j.jenvman.2006.10.010, 2007.

Wu, Z., Dijkstra, P., Koch, G. W., Peñuelas, J. and Hungate, B. A.: Responses of terrestrial

ecosystems to temperature and precipitation change: A meta-analysis of experimental

manipulation, Global Change Biology, 17(2), 927–942, doi:10.1111/j.1365-2486.2010.02302.x, 2011.

Yan, H., Liu, J., Huang, H. Q., Tao, B. and Cao, M.: Assessing the consequence of land use change on agricultural productivity in China, Global and Planetary Change, 67(1), 13–19, doi:10.1016/j.gloplacha.2008.12.012, 2009.

Yang, Y., Fang, J., Ma, W. and Wang, W.: Relationship between variability in aboveground net primary production and precipitation in global grasslands, Geophysical Research Letters, 35(23), L23710, doi:10.1029/2008GL035408, 2008.

Yang, Z.: Inner Mongolia Statistical Bureau, Inner Mongolia statistical yearbook 2002 to 2014 (In Chinese), Mongolian Renmin Press, Hohhot., 2015.

Youhua, R., Xin, L. and Ling., L.: Land Cover Products of China. Cold and Arid Regions Science Data Center at Lanzhou, ., , doi:10.3972/westdc.007.2013.db, 2010.

Yu, D. Y., Shi, P. J., Han, G. Y., Zhu, W. Q., Du, S. Q. and Xun, B.: Forest ecosystem restoration due to a national conservation plan in China, Ecological Engineering, 37(9), 1387–1397, doi:10.1016/j.ecoleng.2011.03.011, 2011.

Zhang, C., Wang, M., Wulanbater and Jiang, X.: Response of ANPP to climate change in Inner Mongolia typical steppes - s simulation study (In Chinese), Acta Botanica Boreali-Occidentalia Sinica, 32(6), 1229–1237, 2012.

Zhang, F., Zhou, G. and Wang, Y.: Dynamics simulation of net primary productivity by a satellite data-driven CASA model in Inner Mongolian typical steppe, China (In Chinese), Journal of Plant Ecology, 32(4), 786–797, doi:10.3773/J.ISSN.1005-264X.2008.04.007, 2008.

Zhang, M., Lal, R., Zhao, Y., Jiang, W. and Chen, Q.: Estimating net primary production of natural

grassland and its spatio-temporal distribution in China, Science of The Total Environment, 553, 184–195, doi:10.1016/j.scitotenv.2016.02.106, 2016.

[revised manuscript text omitted]

**correlation between the number of livestock and the change in the NPP**